# Molecular Fingerprint of BMD Patients Lacking a Portion in the Rod Domain of Dystrophin

**DOI:** 10.3390/ijms23052624

**Published:** 2022-02-27

**Authors:** Daniele Capitanio, Manuela Moriggi, Pietro Barbacini, Enrica Torretta, Isabella Moroni, Flavia Blasevich, Lucia Morandi, Marina Mora, Cecilia Gelfi

**Affiliations:** 1Department of Biomedical Sciences for Health, University of Milan, 20054 Segrate, Italy; daniele.capitanio@unimi.it (D.C.); pietro.barbacini@unimi.it (P.B.); 2Gastroenterology and Digestive Endoscopy Unit, IRCCS Policlinico San Donato, 20097 Milan, Italy; manuela.moriggi@grupposandonato.it; 3IRCCS Istituto Ortopedico Galeazzi, 20161 Milan, Italy; enrica.torretta@grupposandonato.it; 4Child Neurology Unit, Fondazione IRCCS Istituto Neurologico Carlo Besta, 20133 Milan, Italy; isabella.moroni@istituto-besta.it; 5Neuromuscular Diseases and Neuroimmunology Unit, Fondazione IRCCS Istituto Neurologico Carlo Besta, 20133 Milan, Italy; flavia.blasevich@istituto-besta.it (F.B.); luciaovidia.lm@gmail.com (L.M.); moramarinad@gmail.com (M.M.)

**Keywords:** muscle dystrophy, sarcopenia, muscle regeneration, muscle–bone interaction, LC-ESI-MS/MS, 2-D DIGE

## Abstract

BMD is characterized by a marked heterogeneity of gene mutations resulting in many abnormal dystrophin proteins with different expression and residual functions. The smaller dystrophin molecules lacking a portion around exon 48 of the rod domain, named the D8 region, are related to milder phenotypes. The study aimed to determine which proteins might contribute to preserving muscle function in these patients. Patients were subdivided, based on the absence or presence of deletions in the D8 region, into two groups, BMD1 and BMD2. Muscle extracts were analyzed by 2-D DIGE, label-free LC-ESI-MS/MS, and Ingenuity pathway analysis (IPA). Increased levels of proteins typical of fast fibers and of proteins involved in the sarcomere reorganization characterize BMD2. IPA of proteomics datasets indicated in BMD2 prevalence of glycolysis and gluconeogenesis and a correct flux through the TCA cycle enabling them to maintain both metabolism and epithelial adherens junction. A 2-D DIGE analysis revealed an increase of acetylated proteoforms of moonlighting proteins aldolase, enolase, and glyceraldehyde-3-phosphate dehydrogenase that can target the nucleus promoting stem cell recruitment and muscle regeneration. In BMD2, immunoblotting indicated higher levels of myogenin and lower levels of PAX7 and SIRT1/2 associated with a set of proteins identified by proteomics as involved in muscle homeostasis maintenance.

## 1. Introduction

Out-of-frame mutations in the *DMD* gene cause Duchenne muscular dystrophy (DMD), characterized by lack of dystrophin expression and severe phenotype; while in-frame *DMD* mutations cause Becker muscular dystrophy (BMD), with reduced protein expression and milder phenotype. Several intermediate phenotypes between the two forms have also been recognized [1]. While the genetic defect affects mainly skeletal muscle, it may also have additional deleterious effects on the other components of the mechanostat model, such as tendon and bone, as observed in DMD animal models [2,3,4].

Great variability in severity, disease onset, and rate of progression has been observed in BMD patients; this, rather than to the variable amount of dystrophin produced [5], seems more related to the marked heterogeneity of gene mutations that results in many abnormal dystrophin proteins with different expression patterns and residual function [6,7].

Previous studies by us [8,9] had shown that smaller dystrophin molecules lacking a portion in the rod domain were indeed related to milder BMD phenotypes. Subsequently, using a homemade anti-dystrophin monoclonal antibody, called D8, and commercial anti-dystrophin monoclonal antibodies in our routine diagnostic workup, we found that almost one-third of our BMD patients lacked positivity the D8 antibody. The latter was obtained utilizing the fusion peptide containing the portion of dystrophin corresponding to amino acids 2334–2626 (D8 region) of the original dystrophin sequence [10]. From our previous studies, BMD patients were characterized by reduced and uneven dystrophin expression at the muscle fiber surface, assessed by immunohistochemistry, and a reduced amount of dystrophin with expected, or slightly increased MW, detected by SDS-PAGE and immunoblotting, as reported in the literature [8,9]. Conversely, some patients showed normal, or almost normal, dystrophin expression, assessed by immunohistochemistry using commercial monoclonal antibodies, and absence or significantly reduced dystrophin expression, using the D8 monoclonal antibody. Furthermore, SDS-PAGE and immunoblotting showed dystrophin with lower MW and normal or slightly reduced levels. Searching for possible key elements that could help to elucidate which proteins might contribute to improving muscle function in BMD patients, we reanalysed some of the BMD muscle extracts from our recent study on proteomic characterization of BMD and DMD muscle biopsies [11], subdividing them into two groups, BMD1 and BMD2, based on the presence or absence of the D8 antibody reactivity, respectively.

Our working hypothesis was that patients carrying a mutation localized at the rod domain of dystrophin, have a mutated protein that retains its ability to maintain muscle homeostasis. Patients should preserve, at least partially, an almost normal extracellular matrix (ECM) and cytoskeletal structure that compensate for the defect, leading the musculoskeletal-tendon system to “nearly physiologically” counteract fibrosis and dysmetabolism [11,12]. This particular feature could be associable with the ability of these patients to maintain muscle fiber renewal and activation of stem cell recruitment. To explore this possibility in detail, a precise definition of the muscle proteome composition of BMD2 patients is required. The present study explores the molecular signature of BMD2 patients combining top-down in-gel proteomics, label-free LC-ESI-MS/MS, and bioinformatic analysis of all data generated in this and our previous study performed on the same patients [11].

## 2. Results

### 2.1. Immunostaining of Dystrophin Distribution in BMD1 and BMD2 

By immunohistochemistry, we observed two distinct patterns of dystrophin expression, which formed the basis for patient classification into the two groups, BMD1 and BMD2. In BMD1, the typically reduced signal of dystrophin was observed, with irregular and reduced distribution on the sarcolemma. In BMD2, the signal for the D8 portion of dystrophin was absent, while dystrophin appeared almost normally expressed utilising antibodies against other portions of the protein. All components of the dystrophin-associated glycoprotein complex appeared greatly reduced or almost absent in BMD1, and less so in BMD2 (Figure 1).

### 2.2. Proteomic Profiles of Mechanostat Differences in BMD1 and BMD2 

To evaluate changes in protein abundance, vastus lateralis muscle protein extracts were analyzed by liquid chromatography coupled to electrospray tandem mass spectrometry (LC–ESI–MS/MS) with label-free quantification, and by 2-D differential in-gel electrophoresis (2-D DIGE). Overall, the Student’s *t*-test (n = 4, *p*-value < 0.05) revealed 91 changed out of 988 identified proteins in LC–ESI–MS/MS analysis. 2-D DIGE of the same samples identified 50 changed spots between BMD2 and BMD1 and, importantly, highlighted the presence of 38 spots as proteoforms belonging to 15 different proteins, whereas 12 spots were identified as unique proteins. Five proteins were present in both datasets (Figure 2). Identification data for changed proteins/proteoforms are shown in Appendix A for LC–ESI–MS/MS and 2-D DIGE experiments, respectively.

Functional classification of proteins changed in BMD2 vs. BMD1 (Figure 3) revealed a decrease in the extracellular matrix organization (COL6A1, HSPG2, KRT10; a list of abbreviations for identified proteins is provided in Abbreviations) and an overall decrease in cytoskeletal organization and muscle development proteins (ACTA2, CSRP3, FLNC, MAP4, PDLIM3, PLEC, TUBA4A, VIM, FHL1, KLHL40, KLHL41, LGALS1), with some exceptions (increased levels of ACTA1, CDC42, SPTAN1, FHL3 and PHPT1). Muscle contraction fast-twitch type proteins increased (MYH1, MYH2, MYL1, MYLPF, TNNC2, TNNI2, TNNT3), whereas slow-twitch type decreased (MYBPC1, MYH7, MYL3, MYL6B, TNNI1) and sarcomere organization was dysregulated (increased levels of ACTG1, LDB3, MYOZ1, TPM1, and decreased levels of ACTN2, ANKRD2, DYSF, MYOM3, MYOT, MYOZ2, NEB, OBSCN, SYNPO2L, and TTN).

Concerning metabolic proteins, glycolysis and gluconeogenesis proteins increased in BMD2 (ALDOA, ENO3, GAPDH, LDHA, PGK1, PKM), likewise, two enzymes of TCA cycle (CS and DLD) and two aminotransferases (GOT1 and GOT2). Concerning fatty acids beta-oxidation, ECH1, decreased, whereas ECHS1 increased. Mitochondrial respiratory chain and high-energy phosphates conversion components increased (ATP5A1, ATP5B, ATP5C1, ATP5D, ATP5H, ETFA, MT-CO2, NDUFA2, NDUFS3, NDUFV2, UQCRC1, UQCRFS1, ADSSL1, AMPD1, CKM, CKMT2, SLC25A3) with the only exception of ETFB. Immune response was strongly increased (IGHA1, IGHG1, IGKC), whereas the response to oxidative stress was differently dysregulated (increased CA2, HSPA9, PRDX1, PRDX2, and decreased CA3, HSPA1B, HSPB1, HSPB2, PPIA, SOD2). Transport proteins ALB, ANXA6, APOA4, ATP2A2, and TF decreased, while GC, HBA1, and MB increased. Other increased proteins were represented by AHSG, ESD, PHB, SH3BGR, TUFM, YWHAG, YWHAZ, whereas AHNAK, CD59, CRYAB, EEF1A2, HIST1H4A, LMCD1, P4HB, RPLP2, RPSA, were decreased.

### 2.3. Sirtuins Signaling Pathway Revealed by Bioinformatic Analysis

To predict biochemical pathways and functional biological processes related to differentially expressed proteins in muscle extracts of BMD2 vs. BMD1, datasets were processed utilizing the ingenuity pathway analysis (IPA) software. Besides the present dataset, IPA analysis included results from DMD and BMD patients vs. healthy control subjects analysed in a previous study [11]. This combined analysis allowed the identification of canonical pathways, downstream effectors, and upstream regulators that characterize the BMD2 proteomic profile.

The canonical pathway analysis enables recognition of key signaling nodes associated with differentially expressed proteins obtained by proteomic analysis. A total of 5 pathways significantly changed in BMD2 vs. BMD1 (Fisher’s right-tailed exact test *p*-value < 0.05 and z-score ≥ 2 or ≤ −2) were identified. As reported in Table 1, panel A, 4 of them (oxidative phosphorylation, glycolysis, gluconeogenesis, epithelial adherens junction signaling) were predicted to be activated in BMD2 compared to BMD1, while inhibited or not significantly deregulated in DMD and BMD vs. Ctrl. The sirtuin signaling pathway was inhibited in BMD2 vs. BMD1, but slightly activated (although not supported by statistics) both in DMD and BMD vs. Ctrl.

Upstream regulator analysis, performed by IPA software, allowed us to identify upstream factors predicted to regulate groups of proteins changed in our analysis (Table 1, panel B). Among the top upstream regulators controlling the expression of proteins changed in BMD2 vs. BMD1, interleukin-15 (IL15), DEAD-box helicase 5 (DDX5), hypoxia-inducible factor 1 subunit alpha (HIF1A), proteasome activator subunit 3 (PSME3) were activated, while BTB domain and cnc homolog 1 (BACH1), carbonic anhydrase 9 (CA9), and T-cell receptor (TCR) were inhibited.

IPA analysis also enables the prediction of biological functional processes and disorders associated with differentially expressed proteins (Table 1, panel C). In our analysis, the only predicted increased function, based on the z-score, in BMD2 compared to BMD1, was survival of stem cell lines.

### 2.4. Sirtuin Pathway Validation

Sirtuin pathway dysregulation was assessed by analyzing the expression of NAD-dependent protein deacetylases sirtuin-1 (SIRT1) and sirtuin-2 (SIRT2) in our samples. Both SIRT1 and SIRT2 were decreased in BMD2 and BMD1 compared to DMD.

The regenerative capacity of skeletal muscle depends on the availability and proliferative capacity of satellite cells (SCs). The myogenic regulatory transcription factors (MRFs) play a pivotal role in SCs renewal. The expression levels of paired box protein Pax7 (PAX7) and myogenin (MYOG) were assessed by antigen-antibody reaction (Figure 4). PAX7 showed a tendential (not supported by statistics) decrement in BMD2, whereas MYOG was higher in BMD2 compared to BMD1 and DMD.

### 2.5. 2-D DIGE Proteoform Analysis of Moonlighting Proteins

Proteoform analysis of ENO3, ALDOA, and GAPDH was conducted by comparing 2-D DIGE gel images of BMD2, BMD1 and DMD samples (Figure 5a). Concerning ENO3 and GAPDH, proteoform *a* (gel detected pI: pI_ENO3_ = 7.1; pI_GAPDH_ = 7.7) was significantly increased in BMD2 only, while BMD1 and DMD had comparable levels. Proteoforms *b* (gel detected pI: pI_ENO3_ = 7.4; pI_GAPDH_ = 7.9) and *c* (gel detected pI: pI_ENO3_ = 7.7; pI_GAPDH_ = 8.2) progressively increased starting from DMD to BMD1 and BMD2. Proteoform d (gel detected pI: pI_ENO3_ = 8.1; pI_GAPDH_ = 8.4) was increased in BMD2 and BMD1 compared to DMD. ALDOA proteoforms a (gel detected pI = 8.1) and b (gel detected pI = 8.2) progressively increased starting from DMD to BMD1 and BMD2, whereas proteoforms c (gel detected pI = 8.3) and d (gel detected pI = 8.4) increased in BMD2 only (Figure 5b). 

For these proteins, a moonlighting function associated with their possible translocation to the nucleus has been described [13,14,15,16], thus their possible acetylation state indicating the nuclear targeting was assessed by immunoblotting. A qualitative analysis of acetylome performed by 2-D immunoblotting confirmed the acetylated state of ENO3, GAPDH, and ALDOA proteoforms (Figure 5c).

## 3. Discussion

This study refers to a proteomic signature differentiating BMD patients carrying a truncated dystrophin molecule lacking a portion of the rod domain, from BMD patients with normal MW dystrophin and reduced, irregular sarcolemmal expression. D8-negative patients (BMD2) enrolled in this study are at present all too young to manifest possible differences in disease severity. Even so, in these patients, the dystrophin complex is less reduced to the sarcolemma compared to the BMD1, indicating that the smaller dystrophin performs better in anchoring the glycoprotein complex components. Consequently, the subcellular cytoskeleton can be better connected to the extracellular matrix. The BMD2, D8-negative patients, are characterized by the deletion in the dystrophin mRNA of exon 48, in two cases, and deletion of exons 48–51, in the other two cases; such mutations should not delete the spectrin motif repeats R16/17 in the resulting protein and the neuronal nitric oxide (NO) synthase (nNOS) anchoring should be maintained (R16/17 coded by exons 42–45) [17].

In skeletal muscle, nNOS and NO participate in the pathophysiology of muscular dystrophy [18,19]. nNOS is localized to the sarcolemma, facilitates the delivery of NO to the vasculature, and is the only component of the dystrophin complex that is specifically enriched in the sarcolemma of fast-twitch muscle fibers, that preferentially degenerate in Duchenne dystrophy [20]. In BMD patients, the absence of nNOS, but no other components of the dystrophin complex, correlates with the severity of the disease [21]. In the contracting muscle of DMD and BMD patients, the absence of nNOS induces hypoperfusion and ischemia due to vasocostriction. In BMD2, the smaller dystrophin, although slightly reduced, is homogeneously distributed at the sarcolemma. Moreover, a normal expression of nNOS was reported in a 48–51 deleted patient [22] suggesting that in BMD2 patients the nNOS-mediated signalling is retained. Unfortunately, no more tissue was available to verify nNOS expression in our patients.

Results from proteomic analysis of muscle extracts of BMD2 vs. BMD1 indicated a decrease of extracellular matrix and cytoskeletal proteins in the former, suggesting increased muscle stiffness in BMD1, compared to BMD2, that can promote altered muscle tissue homeostasis [23,24,25]. Moreover, protein expression characterizing muscle fiber type indicated increased levels of proteins typical of fast fibers, and proportional decreased level of proteins characterizing slow fibers, in BMD2 compared to BMD1, which, conversely, show the prevalence of proteins characterizing slow-twitch fibers. In addition, BMD2 patients showed increased levels of proteins involved in the sarcomere reorganization. Concerning contractile proteins, proteomics indicated an increase of ACTA2, MYH1, MYH2, MYL1, MYLFP, TNNC2, TNNI2, TNNT3, ACTG1 and TPM1. Transcripts of these proteins were found increased in a previous study addressing the issue of muscle regeneration and stem cell recruitment. The study pinpointed the role of sirtuins inhibition in cell models [26], thus, supporting the hypothesis that BMD2 can synthetize new fast fibers to counteract muscle deterioration.

Cellular energy is generated via glycolysis in the cytoplasm and oxidative phosphorylation in mitochondria. In BMD2 compared to BMD1, muscle metabolic proteins indicated an increase in glycolysis and gluconeogenesis and two enzymes acting as the metabolic node of the TCA cycle (CS and DLD). Concerning the mitochondrial respiratory chain, the majority of enzymes increased, suggesting that the mitochondrial function is more active than in BMD1. The only exception was represented by ETFB levels, which were higher in BMD1. This protein is required for mitochondrial fatty acid oxidation and amino acid metabolism, suggesting that in BMD1 patients, the normal flux of energy production is dysregulated and a metabolic switch toward fatty acid oxidation is expected [27,28]. This metabolic switch induces quiescence of stem cells [26].

Oxidative stress response was at variance with increased levels of proteins controlling the redox system, possibly associated with increased activity of mitochondria and of thioredoxin system (CA2, HSPA9, PRDX1, PRDX2). In contrast, lower levels of CA3, HSPA1B, HSPB1, HSPB2, PPIA, SOD2 were observed.

Among other proteins, interestingly, PHB, a transcriptional co-regulator to the nucleus, and TUFM, a regulator of autophagy and immunity, were increased, whereas ribosomal proteins (RPLP2 and RPSA) and the histone core component of the chromatin (HIST1H4A) decreased, suggesting signaling to the nucleus and chromatin rearrangement [29,30].

IPA pathway analysis, performed on four data sets including BMD2, BMD1, DMD and controls, indicated in BMD2 prevalence of glycolysis and gluconeogenesis and correct flux through the TCA cycle promoting oxidative metabolism, thus supporting the muscle contractile function compared to BMD1 and DMD. These results suggest that the truncated dystrophin molecule lacking a portion of the rod domain can promote positive signaling in muscle tissue, maintaining both metabolism and epithelial adherens junction, while, importantly, the sirtuin signaling appears inhibited. This represents a fundamental aspect for muscle cell renewal since it is well known that the inhibition of SIRT1 promotes histones acetylation and stem cell recruitment [31,32]. Sirtuin signaling was slightly activated in BMD and more active in DMD. The inhibition of sirtuins in BMD2, indicated by IPA analysis, was supported by blotting of SIRT1 and SIRT2 on muscle extracts, which showed, particularly for SIRT2, higher levels in DMD with progressive decrement in BMD1 and a further decrease in BMD2. 

In cancer cells, inhibition of SIRT1/2 has been reported to induce pro-survival autophagy via acetylation of HSPA5 [33]. These two proteins have different targets: SIRT1-mediated forkhead box O3 (FOXO3) and SIRT2-mediated forkhead box O1 (FOXO1) deacetylation; however, both can mediate deacetylation, and their inhibition can promote acetylation of a number of enzymes of the glycolytic pathway [26,33]. Results from in-gel top-down proteome analysis, indicated that specific acidic proteoforms of ALDOA, ENO3 and GAPDH are increased in BMD2 compared to BMD1 and DMD. Furthermore, acetylated proteoforms reflect their increased abundance observed by 2-D DIGE. It is known that glycolytic enzymes, when acetylated, can moonlight to the nucleus and initiate the transcription of several genes [34]. Their nuclear translocation facilitates the delivery of acetyl-CoA used for histone acetylation. This epigenetic modification enables cells to sense the metabolic change and to activate signaling, which, in the case of muscle tissue, consists of the recruitment of satellite cells and initiation of their differentiation [35]. Such event was indeed supported by the higher levels of myogenin and lower levels of PAX7, found in BMD2 patients; and was further corroborated by the disease and biofunction analysis, provided by IPA, utilizing the four datasets described above, indicating that the survival of stem cell lines is increased in BMD2 compared to BMD1, and is inhibited in DMD.

Based on these results, we can speculate that the advantage of BMD2 patients appears to be the activation of epigenetic signaling by translocation of specific acetylated proteoforms of glycolytic enzymes to the nucleus to initiate gene transcription. Unfortunately, due to the limited amount of muscle tissue available for analysis, we could not precisely localize these enzymes. However, this hypothesis is supported by evidence from previous studies. The nuclear localization of aldolase has been described in cardiomyocytes and in smooth muscle cells [13], while enolase was described to translocate from the cytosol to the perinuclear region in muscle under regeneration and in the nucleus of the zona fasciculata of the adrenal cortex [14,36]. Concerning GAPDH, its nuclear translocation is mediated by acetylation of three specific Lys residues (117, 227 and 251) in human cells and the nuclear localization has been associated with DNA damage [15,16].

Dataset analysis highlighted several upstream regulators targeting myogenesis, histone acetylation, oxygen availability and metabolism, and adaptation to oxidative stress, at variance in BMD2 vs. BMD1 and DMD. Interleukin-15 (IL15), DEAD-box helicase 5 (DDX5), hypoxia-inducible factor 1 subunit alpha (HIF1A), proteasome activator subunit 3 (PSME3) were activated, whereas BTB domain and cnc homolog 1 (BACH1), carbonic anhydrase 9 (CA9), and T-cell receptor (TCR) were inhibited in BMD2 vs. BMD1. The activated regulators include the anabolic signaling of IL-15, a chemokine highly expressed and secreted in skeletal muscle, known to promote myogenesis by increasing expression of MyoD and myogenin in differentiated myotubes [37,38]. The activation of DDX5 helicase (p68) targets the chromatin regulatory network coordinating the myogenic differentiation. The complex is located in the myogenin promoter region, where it modulates myogenin expression through post-translational modification (i.e., acetylation) of MyoD, which acts as a key activator of this promoter [39]. 

Another activated regulator is hypoxia and HIF1A signaling. Yang et al. demonstrated the pivotal role of HIFs expression in the self-renewal of satellite cells in hypoxic environments. While in normoxia a knockout of HIF1A/HIF2A did not affect muscle stem cell function, oxygen level < 1% decreased self-renewal and increased differentiation of myoblasts, without effects on cell proliferation [40]. Hypoxia and dysmetabolism are features of muscular dystrophy due either to the genetic defect itself, respiratory insufficiency, or muscle blood vessel abnormalities [41]. Thus, its activation in BMD2 patients could be ascribed to the capacity of their muscles to overcome hypoxia by promoting stem cell recruitment. 

Among inhibited regulatory factors, BACH1 regulating mechanisms involved in ROS production, cell cycle, heme homeostasis, hematopoiesis, and immunity appear to be the most relevant. BACH1 acts by inhibiting the transcription of many oxidative stress-response genes by competing with nuclear factor (erythroid-derived 2)-like-2 (Nrf2) for binding to Maf recognition elements (MAREs) in the gene promoters [42]. A reduction in BACH1 levels suggests increased resistance to oxidative stress and improved redox adaptive homeostasis in BMD2 vs. BMD1. Moreover, the inhibition of TCR-mediated signaling, observed in BMD2, could indicate a reduction of the inflammatory state that follows muscle injury and consequent tissue repair and regeneration [43].

Altogether, these results suggest that, in BMD2 patients, the inhibition of SIRT1/2 promotes stem cell recruitment mediated by glycolytic enzymes and their acetylated proteoforms, which in turn enable muscle differentiation toward fast-twitch fiber renewal. By contrast, BMD1 and DMD, in which a slight activation of sirtuins promotes deacetylation, no changes were observed in the acetylation state (data not shown), the recruitment of satellite cells for muscle renewal is impaired. This hypothesis is supported by high levels of PAX7 and low levels of myogenin, particularly in DMD. However, further experiments will be required to establish which genes are the main targets of this translocation and acetylation. Due to the paucity of the available muscle tissue, we were unable to demonstrate the localization of acetylated proteoforms to the nucleus. Studies are in progress to overcome this issue.

The novelty of the present study is represented by recognition of the role of acetylation of specific glycolytic enzymes targeting the nucleus and the resulting promotion of stem cell recruitment and muscle regeneration in BMD patients carrying dystrophin mutations around exon 48 of the rod domain.

We are aware of the limitations of this study, which involve a restricted number of samples, only partially compensated by the inclusion of proteomics datasets of BMD and DMD obtained adopting the same technologies. The paucity of the tissue, which hampered a nuclear enrichment for the localization of ENO3, GAPDH, and ALDOA, was a major limitation. A validation study on a larger number of samples will be required to confirm our hypothesis; however, the data provided by the present study could contribute to the development of new strategies to support muscle renewal in muscular dystrophies. 

## 4. Materials and Methods

### 4.1. Ethical Statement

Investigations on human tissue were approved by the institutional review board of the Fondazione IRCCS Istituto Neurologico Carlo Besta (Prot. N. 72,74-2006). They were in accordance with Italian law and the Declaration of Helsinki.

### 4.2. Patients and Methods

Vastus lateralis muscle biopsies provided by the Besta neuromuscular biobank (member of Telethon Network of Genetic Biobanks and EuroBioBank) were from eight BMD patients (pts) of our previous study [11]. Specifically, they were patients 18, 19, 20, 22, 24, 28, 29, and 30. Muscle biopsies had been obtained after informed parental consent and frozen in pre-chilled isopentane and stored in liquid nitrogen. In all eight BMD patients, aged 3–9 years, diagnosis had been obtained by dystrophin testing and gene analysis. Patients were subdivided into BMD1 group (pts 18, 19, 20, 24), characterized by reduced and uneven dystrophin expression at the muscle fiber surface, by immunohistochemistry, and of normal molecular weight and reduced band intensity by immunoblotting; and BMD2 group (pts 22, 28, 29, 30), with normal dystrophin expression using commercial monoclonal antibodies, and absent or almost absent signal using the homemade D8 monoclonal antibody, by immunohistochemistry, and of reduced molecular weight, and normal or slightly reduced band intensity by immunoblotting (see Appendix A).

All patients had the muscle biopsy taken because of an occasional finding of hyperCKemia. Clinically, in group 1, patient 18 was asymptomatic at age 15, when last seen; patient 19 showed mild weakness of neck flexors at age 7 years; both this patient and patient 20, unfortunately, were lost to follow-up; patient 24 had developed slight lower limb muscle weakness at age 20 years. In group 2, very mild lower limb girdle muscle weakness was observed in patient 30 at age 14, while patients 22, 28, and 29, were still asymptomatic, respectively, at age 27, 16, and 18 years, when last seen.

Histologically mild myogenic features, consisting of mild variability of muscle fiber diameters, a slight increase of peri and endomysial connective tissue, few centralized nuclei, and rare degenerating fibers, were observed in muscle biopsies in all patients, except in patient 19 that showed normal histological features.

### 4.3. Immunofluorescence Staining

For immunofluorescence, 6-μm-thick cryosections from muscle biopsies of BMD patients and controls were incubated in one of the following primary monoclonal antibodies to dystrophin D8 (homemade, diluted 1:10), dys-2 (Novocastra, Leica Biosystems, Wetzlar, Germany, NCL-DYS2, 1:50), β-sarcoglycan (Novocastra, NCL-b-SARC, 1:10); α-dystroglycan clone IIH6 (Upstate Biotechnology Inc., Lake Placid, NY, USA, 05-593 clone IIH6C4, 1:1); α-dystrobrevin (sc-271630, 1:50), α-syntrophin (sc-166207, 1:10), and sarcospan (sc-393187, 1:10) (all from Santa Cruz Biotechnology, Dallas, TX, USA); followed by incubation in biotinylated anti-mouse IgG or IgM as appropriate (Jackson ImmunoResearch, Ely, UK, 1:250) and in Avidin, NeutrAvidin™, or Rhodamine Red™-Xconjugate (Molecular Probes, Thermo Fisher Scientific Inc., Rockford, IL, USA, 1:250). Muscle sections were examined under a Zeiss Axioplan fluorescence microscope (Carl Zeiss AG, Oberkochen, Germany).

Characterization of the D8 monoclonal, when first obtained in the laboratory, was carried out by testing the antibody specificity on control muscle cryosections following blocking with the peptide against whom it was raised; by observing consistent lack of immunostaining when used in DMD patient muscle and in BMD patients with deletions around exons 48–51; and by verifying that the antibody recognized, on control muscle extracts, a band of the correct 427 kDa molecular weight of dystrophin, which was absent in DMD muscle extracts.

### 4.4. Protein Extraction

Muscle biopsies were divided in two aliquots in a frozen mortar. For label-free proteomics analysis, an aliquot of each frozen muscle was suspended in 2% SDS, 100 mM Tris-HCl pH 7.6, 0.1 M dithiothreitol (DTT), 1 mM phenylmethanesulfonyl fluoride (PMSF) and sonicated on ice until completely dissolved. Lysates were incubated at 95 °C for 3 min and clarified by centrifugation at 16,000× *g* for 5 min at 20 °C. Protein quantitation with Pierce bicinchoninic acid (BCA) protein assay (Thermo Fisher Scientific, Rodano, Italy) was then performed. 

The second aliquot was suspended in DIGE lysis buffer [7 M urea, 2 M thiourea, 4% 3-[(3-cholamidopropyl) dimethylammonio]-1-propanesulfonate (CHAPS), 30 mM Tris, 1 mM PMSF and 20 mM deacetylation inhibition cocktail (Santa Cruz Biotechnology), pH 8.5], and solubilized by sonication on ice. Proteins were selectively precipitated using PlusOne 2D-Clean up kit (GE Healthcare, Little Chalfont, UK) in order to remove non-protein impurities and resuspended in DIGE lysis buffer. Protein extracts were adjusted to pH 8.5 by the addition of 1 M NaOH. Protein concentrations were determined by PlusOne 2D-Quant kit (GE Healthcare).

Sample extracts were analyzed by label-free LC–ESI–MS/MS and 2-D DIGE to evaluate proteome changes. In 2-D DIGE analysis, four DMD patients were added as positive controls (see Appendix A).

### 4.5. Label-Free Liquid Chromatography with Tandem Mass Spectrometry

Protein extracts were processed following the filter-aided sample preparation (FASP) protocol [44,45]. Each sample (200 µg) was deposited in a Microcon-30 kDa centrifugal filter unit (Merck Millipore, Burlington, MA, USA) and washed for two times by centrifugation at 14,000× *g* for 15 min with 200 µL of UA buffer (8 M urea, 0.1 M Tris/HCl, pH 8.5). Samples were carbamydomethylated in 100 µL of 50 mM iodoacetamide in UA buffer for 20 min, then washed for three times in 100 µL UA buffer followed by three washes in 100 µL of 50 mM ammonium bicarbonate in water. Filters were incubated with sequence grade trypsin (Promega, Madison, WI, USA) for 16 h at 37 °C using a protein:trypsin ratio of 50:1. After acidification with trifluoracetic acid and desalting on C18 tips (Zip-Tip C18 micro, Merck Millipore, Burlington, MA, USA), peptide samples were vacuum concentrated, reconstituted in HPLC buffer A (0.1% formic acid) and separated on a Dionex UltiMate 3000 HPLC System with an Easy Spray PepMap RSLC C18 column (250 mm, internal diameter of 75 µm) (Thermo Fisher Scientific, Rodano, Italy), adopting a five steps acetonitrile (ACN)/formic acid gradient (5% ACN in 0.1% formic acid for 5 min, 5–35% ACN in 0.1% formic acid for 139 min, 35–60% ACN in 0.1% formic for 40 min, 60–100% ACN for 1 min, 100% ACN for 10 min, at a flow rate of 0.3 µL/min), and electrosprayed into an Orbitrap Fusion Tribrid (Thermo Fisher Scientific, Rodano, Italy) mass spectrometer. The LTQ-Orbitrap was operated in a positive mode in data-dependent acquisition mode to automatically alternate between a full scan (350–2000 m/z) in the Orbitrap (at resolution 60,000, AGC target 1,000,000) and subsequent CID MS/MS in the linear ion trap of the 20 most intense peaks from full scan (normalized collision energy of 35%, 10 ms activation). Isolation window: 3 Da, unassigned charge states: rejected, charge state 1: rejected, charge states 2+, 3+, 4+: not rejected; dynamic exclusion enabled (60 s, exclusion list size: 200). Mass spectra were analyzed using MaxQuant software (Max-Planck-Institute of Biochemistry, Munich, Germany, version 1.6.3.3). The initial maximum allowed mass deviation was set to 6 ppm for monoisotopic precursor ions and 0.5 Da for MS/MS peaks. Enzyme specificity was set to trypsin/P, and a maximum of two missed cleavages was allowed. Carbamidomethylation was set as a fixed modification, while N-terminal acetylation and methionine oxidation were set as variable modifications. 

The spectra were searched by the Andromeda search engine against the *Homo sapiens* Uniprot UP000005640 sequence database (78,120 proteins, release 7 March 2021). Protein identification required at least one unique or razor peptide per protein group. Quantification in MaxQuant was performed using the built-in extracted ion chromatogram (XIC)-based label-free quantification (LFQ) algorithm using fast LFQ. The required FDR was set to 1% at the peptide, 1% at the protein and 1% at the site-modification level, and the minimum required peptide length was set to seven amino acids. Statistical analyses were performed using the Perseus software (Max Planck Institute of Biochemistry, Munich, Germany, version 1.4.0.6). Each sample was run in triplicate. For each experimental group, the proteins identified in at least 80% of samples were considered. For statistical analysis, a Student’s *t*-test with a *p*-value threshold of 0.05 was applied, and results revealed the variation of protein expression between BMD1 and BMD2 patients. To exclude the presence of false positives from the analysis, Benjamini–Hochberg false discovery rate test was applied. 

### 4.6. Two-Dimensional Differential In-Gel Electrophoresis

Protein labelling, 2-D separation and analysis were performed exactly as previously described [46]. Protein minimal labelling with cyanine dyes (Cy3 and Cy5) was performed, according to manufacturer’s recommendations, by mixing 50 μg of each sample extract with 400 pmol CyDye (GE Healthcare) and incubating, on ice, in the dark for 30 min. The labelling reaction was quenched with 1 mL L-lysine 10 mM on ice for 10 min in the dark. Sample proteins were labelled with Cy5, whereas the internal standard, generated by pooling individual samples (DMD, BMD1, BMD2), was Cy3 labelled. Samples from each subject (40 μg) were combined with an equal amount of internal standard. Each sample was run in triplicate on 24 cm, 3–10 non-linear pH-gradient IPG strips, with a voltage gradient ranging from 200 to 8000 V, for a total of 75,000 Vh, using an IPGphor electrophoresis unit (GE Healthcare). After focusing, proteins were reduced and alkylated. The second dimension was carried out in 20 × 25 cm^2^, 12%T, 2.5% C constant concentration polyacrylamide gels at 20 °C, and 15 mA per gel using the Ettan Dalt II system (GE Healthcare). CyDye-labelled gels were visualized and acquired using a Typhoon 9200 Imager (GE Healthcare). Image analysis was performed using the DeCyder version 6.5 software (GE Healthcare). For each experimental group, spots present in at least 80% of samples were considered. Statistically significant differences of 2-D DIGE data were computed by analysis of variance (ANOVA) and Tukey’s tests (*p* < 0.01). False discovery rate was applied as a multiple test correction in order to keep the overall error rate as low as possible. In case the ANOVA test was not applicable, the non-parametric Kruskal–Wallis test was used. Power analysis was conducted on statistically changed spots, and only spots that reached a sensitivity threshold > 0.8 were considered as differentially expressed. Protein identification was carried out by matrix-assisted laser desorption/ionization–time-of-flight (MALDI-ToF) mass spectrometry (MS). 

For protein identification, semi-preparative gels were loaded with unlabelled sample (400 μg per strip); electrophoretic conditions were the same as 2-D DIGE, and gels were stained with a total-protein fluorescent stain (Krypton, Thermo Fisher Scientific). Image acquisition was performed using a Typhoon 9200 laser scanner. Spots of interest were excised from gel using the Ettan spot picker robotic system (GE Healthcare), destained in 50% methanol/50 mM ammonium bicarbonate, and incubated with 30 μL of 6ng/mL trypsin (Promega) dissolved in 10 mM ammonium bicarbonate for 16 h at 37 °C. Released peptides were subjected to reverse phase chromatography (Zip-Tip C18 micro, Millipore), eluted with 50% acetonitrile (ACN)/0.1% trifluoroacetic acid. Peptides mixture (1 μL) was diluted in an equal volume of 10 mg/mL alpha-cyano-4-hydroxycinnamic acid matrix dissolved in 70% ACN/30% citric acid and processed on an Ultraflex III MALDI-ToF/ToF (Bruker Daltonics, Billerica, MA, USA) mass spectrometer. MS was performed at an accelerating voltage of 20 kV, and spectra were externally calibrated using Peptide Mix calibration mixture (Bruker Daltonics); 1,000 laser shots were taken per spectrum. Spectra were processed by FlexAnalysis software v. 3.0 (Bruker Daltonics) setting the signal to noise threshold value to 6, and search was carried out by correlation of uninterpreted spectra to *Homo sapiens* Uniprot UP000005640 sequence database entries using BioTools v. 3.2 (Bruker Daltonics) interfaced to the on-line MASCOT software, which utilizes a robust probabilistic scoring algorithm. The significance threshold was set at *p*-value < 0.05. No mass and pI constraints were applied and trypsin was set as enzyme. One missed cleavage per peptide was allowed, and carbamidomethylation was set as fixed modification while methionine oxidation as variable modification. Mass tolerance was set at 30 ppm for MS spectra.

### 4.7. Ingenuity Pathway Analysis

Functional and network analyses of statistically significant protein expression changes were performed through Ingenuity Pathway Analysis (IPA) software (Qiagen, Hilden, Germany). In brief, data sets with protein identifiers, statistical test p-values and fold change values calculated from label-free LC-ESI-MS/MS and 2-D DIGE experiments were analyzed by IPA. The “core analysis” function was used to interpret the data through the analysis of biological processes, canonical pathways, upstream transcriptional regulators enriched with differentially regulated proteins. Then the “comparison analysis” function was used to visualize and identify significant proteins or regulators across experimental conditions. The present dataset was compared with previous results obtained by comparing DMD and BMD patients with healthy control subjects [11]. *p*-values were calculated using a right-tailed Fisher’s exact test. The activation z-score was used to predict the activation/inhibition of a pathway/function/regulator [47]. A Fisher’s exact test *p*-value < 0.05 and a z-score ≤ −2 and ≥ 2, which takes into account the directionality of the effect observed, were considered statistically significant.

### 4.8. Acetylome Analysis

Qualitative analysis of acetylated proteoforms was conducted on 2-D DIGE total protein extracts. 2-D immunoblotting was carried out on DMD, BMD1 and BMD2 pooled samples by subjecting each pool (150 μg) to isoelectrofocusing on 18 cm, 6–10 pH-gradient IPG strips (GE Healthcare), with a voltage gradient ranging from 300 to 8000 V, for a total of 52,000 Vh, using an IPGphor electrophoresis unit (GE Healthcare). After focusing, proteins were reduced and alkylated. The second dimension was carried out in 20 × 25 cm^2^, 12% polyacrylamide gels at 20 °C. Blots were blocked in 5% BSA for 30 min and incubated with a 1:1 mixture of rabbit anti-acetylated-lysine (Ac-K2–100) (1:1000, Cell Signaling Technology, Danvers, MA, USA, #9814) and anti-acetylated-lysine (1:1000, Cell Signaling Technology, #9441) primary antibodies. After washing, membranes were incubated with anti-rabbit HRP-conjugated secondary antibody (1:10,000, GE Healthcare). Signals were visualized by chemiluminescence using the ECL Prime (GE Healthcare) detection kit and the Image Quant LAS 4000 (GE Healthcare) analysis system.

### 4.9. Immunoblotting

Protein extracts (50 µg) from DMD, BMD1, and BMD2 vastus lateralis muscle samples were loaded and resolved on 10% (PAX7 and MYOG) and 8–12% gradient (SIRT1/2) polyacrylamide gels. Blots were incubated with mouse monoclonal anti-Pax7 (Santa Cruz Biotechnology, Dallas, TX, USA, sc-81648, 1:500), rabbit polyclonal anti-myogenin (Santa Cruz Biotechnology, Dallas, TX, USA, sc-576, 1:500), rabbit monoclonal anti-SirT1 (Cell Signaling Technology, Danvers, MA, USA, #2496, 1:1000) and anti-SirT2 (Cell Signaling Technology, #12650, 1:1000). After washing, membranes were incubated with anti-rabbit (GE Healthcare, 1:10,000) or anti-mouse (Jackson ImmunoResearch, Ely, UK, 1:5000) secondary antibody conjugated with horseradish peroxidase. Signals were visualized by chemiluminescence using the ECL Prime detection kit and the Image Quant LAS 4000 (GE Healthcare) analysis system. Band quantification was performed using the Image Quant TL (GE Healthcare) software followed by statistical analysis (ANOVA + Tukey, n = 4, *p*-value < 0.05). Band intensities were normalized against the total amount of proteins stained by Sypro ruby total-protein stain. Full-length images are available in Appendix A.

## Figures and Tables

**Figure 1 ijms-23-02624-f001:**
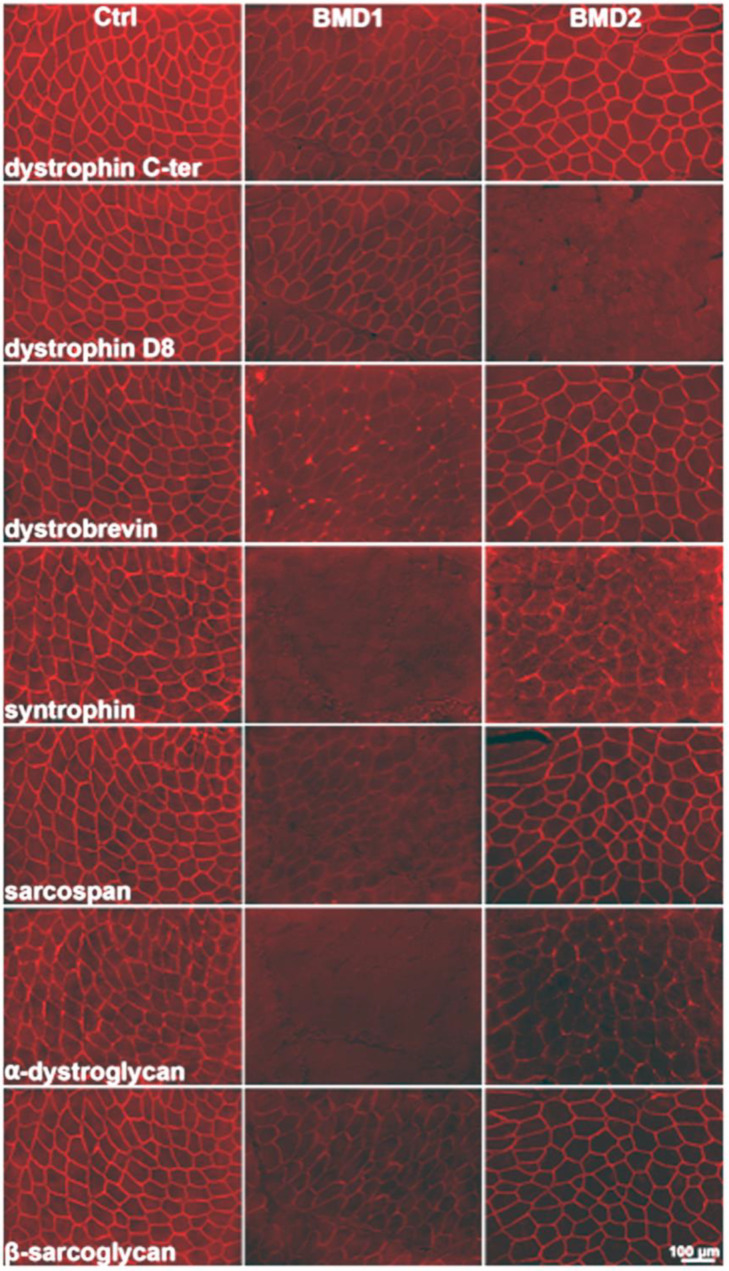
Immunostaining of dystrophin (detected using antibodies against the C-terminal and D8 regions) and dystrophin-associated proteins (alpha-dystrobrevin, alpha-syntrophin, sarcospan, alpha-dystroglycan, and beta-sarcoglycan), in consecutive sections of *Vastus lateralis* muscle from a control (**left** column of the panel), a BMD1 patient **(central** column), and a BMD2 patient (**right** column).

**Figure 2 ijms-23-02624-f002:**
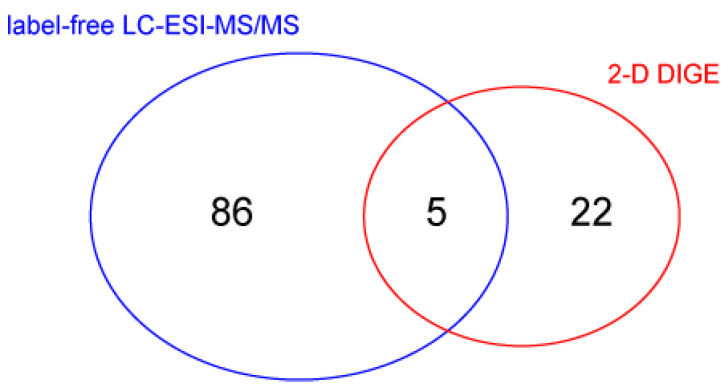
Venn diagram showing the number of identified proteins changed in BMD2 vs. BMD1 group, as detected with label-free LC-ESI-MS/MS and 2-D DIGE proteomic approaches.

**Figure 3 ijms-23-02624-f003:**
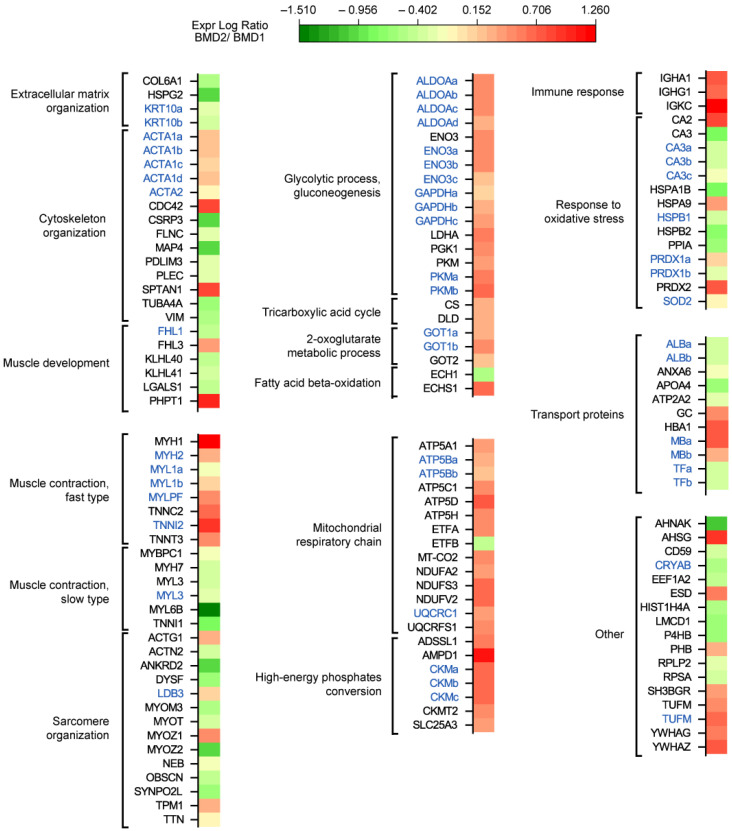
Heatmap of protein expression data divided according to functional categories. Green and red colors refer to statistically significant decrease or increase for each individual protein/proteoform (BMD2 vs. BMD1, Student’s *t*-test and FDR, n = 4, *p* < 0.05) in our proteomics datasets (black labels, Label-free LC-ESI-MS/MS; blue labels, 2-D DIGE). Proteoforms are identified with a lowercase letter, in alphabetical order based on their pI, from more to less acidic. The list of abbreviations for identified proteins is provided in Abbreviations.

**Figure 4 ijms-23-02624-f004:**
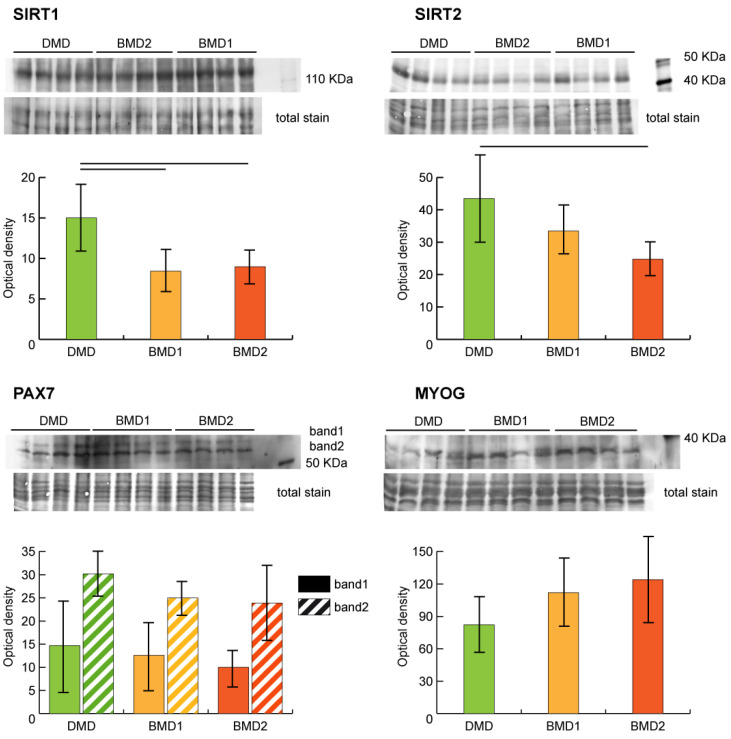
Representative immunoblot images and bar graphs (mean ± SD) showing protein abundance of NAD-dependent protein deacetylases sirtuin-1 (SIRT1) and sirtuin-2 (SIRT2), paired box protein Pax7 (PAX7), and myogenin (MYOG). Data were normalized against the total amount of loaded proteins stained with Sypro Ruby. Statistical analysis was performed by one-way ANOVA and Tukey’s test (horizontal black lines above the bars = significant changes, n = 4, *p* < 0.05). Full-length images are available in Appendix A.

**Figure 5 ijms-23-02624-f005:**
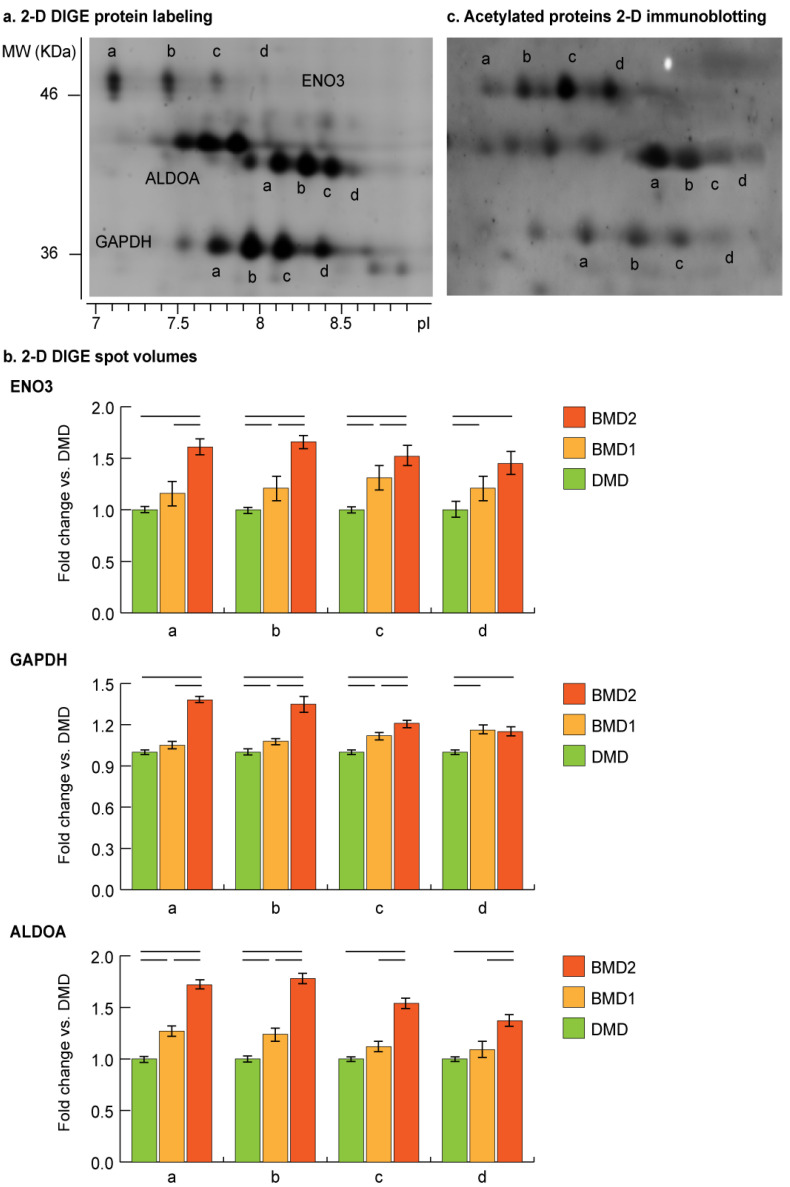
(**a**) Representative close-up image showing beta enolase (ENO3), glyceraldehyde-3-phosphate dehydrogenase (GAPDH) and fructose-bisphosphate aldolase A (ALDOA) proteoforms distribution in a 2-D DIGE gel. Proteoforms were labeled in alphabetical order according to their pI, from more to less acidic. (**b**) 2-D DIGE proteoforms relative quantitation. Statistical analysis was performed by one-way ANOVA and Tukey’s test (horizontal black lines above the bars = significant changes, n = 4, *p* < 0.05). (**c**) Representative close-up image of the same region after 2-D blotting and immunostaining with an anti-acetylated-lysine antibody. Corresponding proteoforms are marked with the same letter.

**Table 1 ijms-23-02624-t001:** Canonical pathways (**a**), upstream regulators (**b**), and diseases and biofunction (**c**) heatmaps display the most significant results (ordered by decreasing z-scores in BMD2 vs. BMD1) resulting from an IPA comparison analysis across different datasets. In particular, results from the present analysis (first column) are compared with data elaborated from a list of differentially changed proteins obtained by comparing DMD and BMD patients with healthy controls adopting the same methodologies [11]. The orange and blue-colored rectangles indicate predicted pathway activation or predicted inhibition, respectively, via the z-score statistic (significant z-scores ≥ 2, ≤ −2).

**a. Canonical Pathways**	**BMD2/BMD1**	**DMD/BMD**	**DMD/ctrl**	**BMD/ctrl**
Oxidative Phosphorylation	3.317	−2.646	−0.707	1.134
Glycolysis I	2.236	−2.236	−3.317	−2.828
Gluconeogenesis I	2	−2.236	−2.53	−1.633
Epithelial Adherens Junction Signaling	2	N/A	1.633	N/A
Sirtuin Signaling Pathway	−2.121	0.707	0.905	1.414
**b. Upstream Regulators**	**BMD2/BMD1**	**DMD/BMD**	**DMD/ctrl**	**BMD/ctrl**
IL15	2.438	−2.438	−1.874	−2.091
DDX5	2.236	N/A	0	1.342
HIF1A	2.099	0.014	-0.226	−1.413
PSME3	2	−2	−1.89	−1.633
BACH1	−2.213	N/A	N/A	N/A
CA9	−2.449	2.236	1	−1
TCR	−2.474	1.373	0.511	−1.265
**c. Diseases and Bio Functions**	**BMD2/BMD1**	**DMD/BMD**	**DMD/ctrl**	**BMD/ctrl**
Survival of stem cell lines	2.236	N/A	−1.134	0

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
