# Peer review of "Molecular Fingerprint of BMD Patients Lacking a Portion in the Rod Domain of Dystrophin"

_ijms, 2022, doi:10.3390/ijms23052624_

Round 1

Reviewer 1 Report

This manuscript describes ex vivo data about how the proteome profile of the vastus lateralis muscle is altered in a subgroup of patients with Becker muscular dystrophy. The data presented here is clinically relevant and the manuscript is well written. However, a few points should be clarified before the final publication of the manuscript.

Specific suggestions:

  1. Subtitles of Results should imply the obtained most important conclusions.
  2. In the legend of Figure 1, it should be included from which muscle the biopsies were taken.
  3. The statistically significant changes are not shown in Figure 4. Was One-way ANOVA used in this case? Please, clarify.
  4. Figure 4 contains bar graphs instead of histograms.
  5. Statistical analysis was not performed and described in Figure 5.
  6. References should be included to the following statements: lanes 277-281, 288-290, 305-307, and 341-343.
  7. Catalog numbers of the commercially available antibodies should be included in 4.3.
  8. All abbreviations should be given when first mentioned in the text.
  9. There are minor grammatical or spelling errors in lanes 43, 64, 102, 140, 150, and 261.

Author Response

Reviewer 1

Comments and Suggestions for Authors

This manuscript describes ex vivo data about how the proteome profile of the vastus lateralis muscle is altered in a subgroup of patients with Becker muscular dystrophy. The data presented here is clinically relevant and the manuscript is well written. However, a few points should be clarified before the final publication of the manuscript.

Specific suggestions:

  1. Subtitles of Results should imply the obtained most important conclusions.
  2. In the legend of Figure 1, it should be included from which muscle the biopsies were taken.
  3. The statistically significant changes are not shown in Figure 4. Was One-way ANOVA used in this case? Please, clarify.
  4. Figure 4 contains bar graphs instead of histograms.
  5. Statistical analysis was not performed and described in Figure 5.
  6. References should be included to the following statements: lanes 277-281, 288-290, 305-307, and 341-343.
  7. Catalog numbers of the commercially available antibodies should be included in 4.3.
  8. All abbreviations should be given when first mentioned in the text.
  9. There are minor grammatical or spelling errors in lanes 43, 64, 102, 140, 150, and 261.

Submission Date

31 January 2022

Date of this review

06 Feb 2022 14:46:33

Reply to reviewer 1:

The authors would like to thank the Reviewer for his valuable contribution.

1. Subtitles of Results should imply the obtained most important conclusions.
R. Subtitles have been rewritten according to Reviewer’s suggestions.

2. In the legend of Figure 1, it should be included from which muscle the biopsies were taken.
R. Vastus lateralis muscle has been added to the legend.

3. The statistically significant changes are not shown in Figure 4. Was One-way ANOVA used in this case? Please, clarify.
R. Figure 4 has been modified and the statistical method explained in the caption.

4. Figure 4 contains bar graphs instead of histograms.
R. The term “histograms” has been changed to “bar graphs”.

5. Statistical analysis was not performed and described in Figure 5.
R. We apologize for the oversight, statistical analysis has been described in the legend.

6. References should be included to the following statements: lanes 277-281, 288-290, 305-307, and 341-343.
R. The authors are grateful for this suggestion, references have been added.

7. Catalog numbers of the commercially available antibodies should be included in 4.3.
R. Catalog numbers have been added in the section.

8. All abbreviations should be given when first mentioned in the text.
R. The authors agree with the Reviewer, however, regarding protein abbreviations, these have been included in full in Appendix A to make the text more fluent for the reader.

9. There are minor grammatical or spelling errors in lanes 43, 64, 102, 140, 150, and 261.
R. The authors thank the Reviewer for highlighting spelling errors, which have now been corrected.

Reviewer 2 Report

The authors have classified the BMD phenotypes into two groups based on the D8 antibody specificity (D8 region of Dystrophin) and performed a detailed proteomics analysis on it. The study is almost complete and the conclusion drawn are well supported by the reult. I have few minor comments

  1. In Fig 1, the authors should have used a blocking peptide of D8 to show the specificity of D8 antibody.
  2. The proteomics study was performed with whole tissue rather than the purified muscle cells. The authors should discuss about this limitation in the discussion section.
  3. In Fig 4, the authors should show the loading control.

Author Response

Reviewer 2

Comments and Suggestions for Authors

The authors have classified the BMD phenotypes into two groups based on the D8 antibody specificity (D8 region of Dystrophin) and performed a detailed proteomics analysis on it. The study is almost complete and the conclusion drawn are well supported by the reult. I have few minor comments

  1. In Fig 1, the authors should have used a blocking peptide of D8 to show the specificity of D8 antibody.
  2. The proteomics study was performed with whole tissue rather than the purified muscle cells. The authors should discuss about this limitation in the discussion section.
  3. In Fig 4, the authors should show the loading control.

Submission Date

31 January 2022

Date of this review

12 Feb 2022 23:33:43

Reply to reviewer 2:

The authors thank the Reviewer for suggestions aimed to improve the quality of the article.

1. In Fig 1, the authors should have used a blocking peptide of D8 to show the specificity of D8 antibody.

R. The authors thank the Reviewer for this important observation. A paragraph has been added in section 4.3 detailing how the specificity of the antibody was assessed.

2. The proteomics study was performed with whole tissue rather than the purified muscle cells. The authors should discuss about this limitation in the discussion section.
R. The major limitation in the use of patient muscle biopsies is the amount of sample available for experiments. Specifically, in our case, muscle biopsies were retrieved from a biobank where they had been stored frozen for clinical purposes. Therefore, it was not possible to obtain muscle cell cultures from frozen material.

3. In Fig 4, the authors should show the loading control.
R. Loading control has been added in the figure.

Round 2

Reviewer 1 Report

The manuscript was substantially improved. Therefore, I suggest the current version of the manuscript for publication in International Journal of Molecular Sciences.